# No Need to Talk:
# Asynchronous Mixture of Language Models

**Anastasiia Filippova** [†]
EPFL

**Angelos Katharopoulos**
Apple

**David Grangier**
Apple

**Ronan Collobert**
Apple

## Abstract

We introduce SMALLTALK LM, an innovative method for training a mixture of language models in an almost asynchronous manner. Each model of the mixture specializes in distinct parts of the data distribution, without the need of high-bandwidth communication between the nodes training each model. At inference, a lightweight router directs a given sequence to a single expert, according to a short prefix. This inference scheme naturally uses a fraction of the parameters from the overall mixture model. Unlike prior works on asynchronous LLM training, our routing method does not rely on full corpus clustering or access to metadata, making it more suitable for real-world applications. Our experiments on language modeling demonstrate that SMALLTALK LM achieves significantly lower perplexity than dense model baselines for the same total training FLOPs and an almost identical inference cost. Finally, in our downstream evaluations we outperform the dense baseline on 75% of the tasks.

## 1 Introduction

Recent research has demonstrated that scaling large language models (LLMs) by increasing model capacity and expanding training data consistently leads to significant performance improvements on a wide range of downstream tasks (Kaplan et al., 2020; Brown et al., 2020; Henighan et al., 2020; Hoffmann et al., 2022; Dubey et al., 2024). Scaling introduces substantial operating and engineering costs for both inference and training. In general, training is achieved on a large number of nodes via synchronous gradient descent techniques, which relies on high-bandwidth communication to scale. Inference of large models may require multiple compute nodes to distribute the model, which relies on low-latency communication. In both cases, state-of-the-art interconnect hardware is critical, and careful engineering is required to scale, and maintain a large compute cluster. While mainstream machine learning frameworks have eased the engineering work on the scaling side, access to a large number of well interconnected nodes remains a privilege in the machine learning community.

In this paper, we explore strategies to mitigate the communication cost of large language models, both at training and inference, while keeping the inference efficient. We show that efficient training and inference can be achieved without relying on fast interconnects, and without compromising model performance, both in terms of perplexity or downstream task accuracy.

In recent studies aimed at mitigating reliance on high-bandwidth interconnects, researchers have developed algorithms that reduce the need for frequent or comprehensive gradient synchronizations. Such techniques include asynchronous training (Douillard et al., 2023; Aji & Heafield, 2017; Zhang et al., 2015; Liu et al., 2024) and gradient compression (Lin et al., 2017; Dettmers, 2016) which effectively decrease communication overhead. By performing updates less frequently or by communicating less data, these methods sustain high training throughput while diminishing dependence on high-speed interconnects. However, these algorithms still require some level of gradient synchronization, and resulting models often under perform (in terms of perplexity) compared to training approaches synchronizing at every step (Diskin et al., 2021).

Regarding efficient inference, a number of sparse parameter activation techniques have recently become popular (Shazeer et al., 2017; Fedus et al., 2022; Artetxe et al., 2022; Lewis et al., 2021; Du et al., 2022), in particular the Switch Transformer mixture of experts (MoE). These approaches

---

[†]Work done while interning at Apple.

are effective at reducing significantly the number of active model parameters at inference time. Yet, they require routing decisions to be made for each token, demanding rapid interconnections and requiring all parameters to be accessible in RAM.

To achieve both low-bandwidth training and efficient inference, several prior works have explored asynchronously training mixtures of models (Gross et al., 2017; Li et al., 2022; Gururangan et al., 2023). In these methods, the input space is clustered using unsupervised techniques like K-Means, with each model in the mixture trained on an assigned cluster. These approaches reduce communication overhead and have demonstrated improvements in model perplexity, while maintaining similar inference costs to baseline models (Gross et al., 2017; Gururangan et al., 2023). However, in the context of language processing, their practical applicability on standard downstream tasks with relatively short input prefixes remains a subject of debate, as they rely on full corpus clustering or access to metadata (Li et al., 2022; Gururangan et al., 2023).

In this paper, we introduce SMALLTALK LM, an asynchronous mixture of language models, which combines the advantage of asynchronous LM training methods (significantly reducing interconnect requirements), and sparse activation methods (it naturally uses only a subset of the overall architecture parameters during inference). SMALLTALK LM achieves better perplexity, and better accuracy on a majority of downstream tasks, compared to a regular dense language model trained with the same amount of FLOPs. Our main contributions are as follows:

- We introduce an algorithm that trains a mixture of independent language models, significantly reducing bandwidth requirements during training compared to regular distributed training methods. Our approach leverages a lightweight router (which accounts for less than $1.5\%$ of the size of each expert) to efficiently route sequences. This router determines the most suitable expert for a given sequence based on a short prefix (see § 2).

- We empirically demonstrate that our method achieves significantly lower perplexity than a dense baseline for near-identical training and inference FLOPs and identical training data volume (see Fig. 2). Furthermore, we show that for a constant mixture model size, the improvement in perplexity increases with the number of experts.

- We evaluate our method on a set of downstream tasks, showing that it achieves better or similar accuracy compared to a dense baseline on $75\%$ of the tasks (see § 3.3 and App. B).

## 2 METHOD

In this section, we formalize our proposed method: SMALLTALK LM. In § 2.1, we introduce general language modeling background, and present a mixture of experts framework in this context. Subsequently, in § 2.2, we show how we can use independent language models to perform routing to different experts in the mixture. We also present the training procedure and implementation details of our method. Finally, we discuss its marginal computational overhead and minimal distributed communication costs.

### 2.1 BACKGROUND

**Language modeling**  Let $\mathcal{V}$ denote a fixed vocabulary, and consider a sequence of tokens $\mathbf{x} = (x_1, x_2, \ldots, x_S)$, where each $x_s \in \mathcal{V}$ for $s = 1, 2, \ldots, S$. Language modeling is about modeling the data distribution $p(\mathbf{x})$, which in practice is cast as training a language model $p(x_{s+1} \mid \mathbf{x}_{1:s} ; \theta)$ generating a token $x_{s+1}$ given a context sequence $\mathbf{x}_{1:s} = (x_1, x_2, \ldots, x_s)$. Typically, the model parameterized by $\theta$ is a neural network, and parameters are obtained by minimizing the negative log-likelihood:

$$\mathcal{L}(\mathbf{x}; \theta) = -p(\mathbf{x} ; \theta) = -\sum_{s=1}^{S-1} \log p(x_{s+1} \mid \mathbf{x}_{1:s} ; \theta) \,. \tag{1}$$

**Mixture of Experts**  There exist many variants of the mixture of experts model (Jacobs et al., 1991; Jordan & Jacobs, 1994; Tresp, 2000; Collobert et al., 2001; Shahbaba & Neal, 2009). Following the original formulation and applying it to language modeling, we factorize the conditional distribution for predicting a given token by summing over a set of $E$ experts:

$$\mathcal{L}(\mathbf{x};\theta) = -\sum_{s=1}^{S-1} \log \sum_{e=1}^{E} p(x_{s+1} \mid \mathbf{x}_{1:s}, e\,;\,\theta^e)\, p(e \mid \mathbf{x}_{1:s}\,;\,\theta^r), \tag{2}$$

where each expert $p(x_{s+1} \mid \mathbf{x}_{1:s}, e\,;\,\theta^e)$ is intended to focus on different parts of the data distribution, and the router $p(e \mid \mathbf{x}_{1:s}\,;\,\theta^r)$ assigns weights to the expert predictions. Each expert $e$ has its own set of independent parameters $\theta^e$, and the router is parameterized by $\theta^r$.

While this type of mixture of experts partitions the input space across different models, all experts are still involved when computing the conditional likelihood in Equation (2), both at training and inference time. Hard mixtures of experts circumvent this issue by hard-assigning a sequence $\mathbf{x}_{1:s}$ to a *single* expert $e^\star$, minimizing an upper bound of Equation (2):

$$\mathcal{L}(\mathbf{x};\theta) \le -\sum_{s=1}^{S-1} \log p(x_{s+1} \mid \mathbf{x}_{1:s}, e^\star\,;\,\theta^{e^\star})\, p(e^\star \mid \mathbf{x}_{1:s}\,;\,\theta^r), \tag{3}$$

The way $e^\star$ is chosen depends of the goal of the approach. For example, in Jacobs et al. (1991), $e^\star$ is sampled at training time, according to the distribution $p(e \mid \mathbf{x}_{1:s}\,;\,\theta^r)$. In other works, like Collobert et al. (2001), it is chosen as the most relevant expert (that is $e^\star = \arg\max_e p(e \mid \mathbf{x}_{1:s}\,;\,\theta^r)$), both at training and inference. Regarding the training procedure, mixture models with both soft and hard assignments are typically trained with an Expectation-Minimization (EM) scheme, alternating between experts and router optimization.

## 2.2 SMALLTALK LM

**Routing With Independent Language Models**   SMALLTALK LM can be viewed as an instance of hard mixture of experts, where a sequence is routed to the most relevant expert $e^\star$, according to the router posterior $p(e \mid \mathbf{x}_{1:s}\,;\,\theta^r)$, both at training and inference. However, instead of being a monolithic neural network model, the router is implemented as a language model per expert. More precisely, with Bayes' rule we have:

$$e^\star = \arg\max_e p(e \mid \mathbf{x}_{1:s}\,;\,\theta^r) \tag{4}$$

$$= \arg\max_e \frac{p(\mathbf{x}_{1:s} \mid e\,;\,\theta^r)\,p(e)}{\sum_{i=1}^{E} p(\mathbf{x}_{1:s} \mid i\,;\,\theta^r)\,p(i)} \tag{5}$$

$$= \arg\max_e p(\mathbf{x}_{1:s} \mid e\,;\,\theta^r) \tag{6}$$

$$= \arg\max_e p(\mathbf{x}_{1:s} \mid \theta^{r,e})\,, \tag{7}$$

where the likelihood $p(\mathbf{x}_{1:s} \mid \theta^{r,e})$ defines a routing language model attributed to the $e$-th expert with *independent* parameters $\theta^{r,e}$, and expert priors $p(e)$ are supposed to be uniform. So far, in our definition, there is nothing that prevents $\theta^{r,e} = \theta^e$, namely to use the experts themselves to implement the router. Interestingly, in that case selecting $e^\star$ as above is intuitive because it amounts to selecting the best expert for a given sequence (in terms of log-likelihood).

**Deriving a Practical Model**   In order for the above formulation to result in a model that is useful in practice, we have to solve the following main challenges. Firstly, we need to be able to route using a small prefix of the full sequence since in most real world applications we do not have access to a full sequence (e.g. in question answering we do not have the answer). Secondly, using the experts to implement the router results in a large computational overhead since we need to evaluate every single one of them, largely negating the benefits of the hard mixture of experts.

We solve both of the above problems with the following two key ideas:

1. We use a *short prefix* of length $M$ to compute the score for each expert as follows:

$$p(e^\star \mid \mathbf{x}_{1:s}) \approx p(e^\star \mid \mathbf{x}_{1:M}). \tag{8}$$

This enables us to use SMALLTALK LM in real-world applications where only a very small prefix may be available. We show in § 3.4 that our method outperforms the dense baseline even when using as little as 32 tokens for routing.

2. We implement the router with $E$ language models that are *orders of magnitude smaller* than the experts. This results in marginal computational overhead, as little as $1\%$ during training and $3\%$ during inference (§ 3.2).

Using the log-likelihood of a prefix to route a sequence is an intuitive solution since a model that is good at predicting the beginning of a sequence is likely to be good at the full sequence. Another interpretation is that the prefix can be used to perform some sort of thematic identification of the whole sequence that is then used to route to the appropriate expert.

On the other hand, the efficacy of smaller language models as routers is less intuitive. Even though one can expect a large model to be able to perform well on sequences that are beyond the capabilities of smaller models, the task of routing boils down to identifying whether a sequence is similar to the training data or not. We show that even really small models ($4.4$M parameters) can perform that task very well and result in no discernible difference in performance compared to larger routers (§ 3.4).

---

**Algorithm 1 SMALLTALK LM training.**

---

1: Train the routers
2: $\mathbf{X} \leftarrow N$ new sequences from the dataset
3: $\mathbf{X}_{1:E} = \text{random\_assignments}(\mathbf{X})$
4: **for** $i = 1 \ldots T$ **do**
5:     **for** $e = 1 \ldots E$ **do**
6:         $\theta^{r,e} \approx \arg\min_{\theta^{r,e}} \mathcal{L}(\mathbf{X}_e; \theta^{r,e})$       # Optimize Equation 9 with SGD for the $e$-th router
7:     **end for**
8:     $\mathbf{X} \leftarrow N$ new sequences from the dataset
9:     $\mathbf{X}_{1:E} = \text{balanced\_assignments}(\mathbf{X}, \theta^r)$       # Segment the data according to Equation 4
10: **end for**

11: Train the experts
12: $\mathbf{X} \leftarrow M$ new sequences from the dataset comprising the total number of training tokens
13: $\mathbf{X}_{1:E} = \text{balanced\_assignments}(\mathbf{X}, \theta^r)$
14: **for** $e = 1 \ldots E$ **do**
15:     $\theta^e \approx \arg\min_{\theta^e} \mathcal{L}(\mathbf{X}_e; \theta^e)$       # Optimize Equation 1 with SGD for the $e$-th expert
16: **end for**

---

**Training Procedure**    We train our proposed hard mixture of experts with EM. Namely we alternate between a step of optimizing equation 1 and a step of assignments according to equation 4. In our framework, assignments $e^\star$ purposefully do not depend on the experts, and the routers are independent language models. This allows us to split the training in two stages:

1. We train the routers using EM without using the experts at all, namely we minimize

$$\mathcal{L}(\mathbf{x}, \theta^r) = -\log p(e^\star \mid \mathbf{x}_{1:M}; \theta^r) = -\sum_{s=1}^{M-1} \log p\left(x_{s+1} | \mathbf{x}_{1:s}; \theta^{r,e^\star}\right), \qquad (9)$$

in the likelihood maximization step and perform the assignments using equation 4 in the expectation step.

2. We train the experts as $E$ independent language models on disjoint dataset segments selected by the trained routers.

This procedure is described in pseudocode in Algorithm 1. In the specific case where the experts and the routers are the same language models ($\theta^e = \theta^{r,e}$) our training algorithm comes down to a regular EM algorithm where the sequences are routed to the best performing expert. In that light, we can think of each router as lightweight approximation of the corresponding expert. See § 3.4 for experiments where the experts themselves are used for routing.

**Balancing the Assignments to Experts**    A common hurdle to overcome when training mixture of experts models is to ensure that the modeling load is shared as evenly as possible across all experts, or in other words that the model is utilizing the capacity of all experts efficiently instead of relying on a few good ones. Common ways of tackling this problem are either introducing losses that penalize

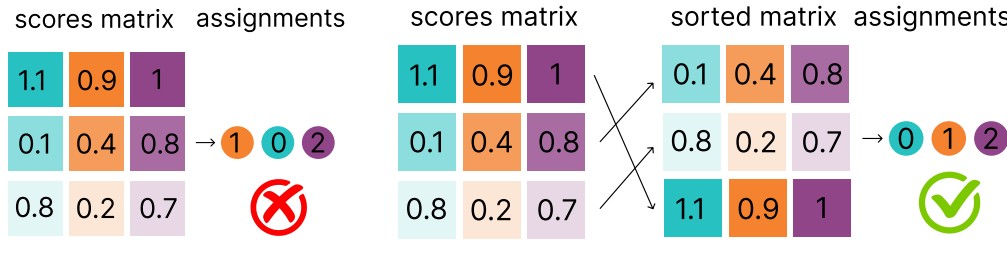

(a) Naive assignments    (b) Assignment with respect to the minimum in a chunk.

Figure 1: **Balanced assignments.** In this scenario we have 3 experts (columns with different colors) and 3 sequences to assign (rows) under the constraint that each expert should get 1 sequence. In **(a)** we assign sequentially each row, by negative log-likelihood, leading to sub-optimal assignment because the first expert is full when we try to assign the last row. On the other hand, in **(b)**, we first sort wrt to the minimum log-likelihood which results in optimal assignments.

uneven assignments or introducing noise into the assignments namely by sometimes performing "sub-optimal" assignments to avoid local-minima. These techniques are used in all mixture models since their introduction in Jacobs et al. (1991) until more recently in Fedus et al. (2022).

In our setup, we can solve this problem by ensuring that each expert is assigned equally sized distinct portions of the dataset. However, a challenge arises when assigning data sequentially based solely on Equation 4. In such cases, an expert might already have more than its fair share of the dataset assigned yet we continue to find sequences for which it has the lowest loss among all experts, as illustrated in Figure 1a. To overcome this issue, during training we perform the assignments considering the whole set of sequences rather than one sequence at a time. In particular, we sort the sequences based on $-\max_e \log p(\mathbf{x}_{1:M} \mid e; \theta^r)$ and perform the assignments in that order. This ensures that sequences with high likelihood will be assigned first and avoids the edge-case where a sequence with low-loss cannot be assigned to the appropriate expert because it is at capacity. In Algorithm 1 we refer to this procedure as *balanced assignments* and it is demonstrated in Figure 1b.

During inference, no balancing is performed, and the expert is selected solely based on equation 4.

**Computational and Communication Cost**   It is important to note that the computational cost of training and inference with the routers is significantly lower than that of training and inference with the experts. This difference arises due to the routers' much smaller size, smaller batch sizes, and the fewer tokens they train on (see Appendix A.3 for further details). As a result, the additional computational overhead from training and inference with the routers is negligible.

Regarding communication requirements for a distributed implementation, in Algorithm 1 all operations except for the balanced assignments are completely independent and embarrassingly parallelizable without any communication. Namely, each expert or each router can be trained on its own on a different node or set of nodes. The assignments, however, require that every node has access to the scores of every router in order to perform the assignment according to equation 4. This requires a small amount of communication as each node needs to share 1 score per sequence which results in 2MB per 1M sequences if we represent the scores in 16-bit floats. A detailed analysis is provided in the Appendix A.4.

## 3 EXPERIMENTS

In this section, we experimentally analyze the performance of our proposed method. First, in § 3.2 we benchmark our approach against a dense LLM trained using regular distributed training techniques. Second, in § 3.3, we evaluate our method on downstream tasks, demonstrating that it achieves performance comparable to baseline models at the same level of perplexity, improving upon a dense model given the total training FLOPs spent. Finally, in § 3.4, we analyse the core component of our method, the routing. We show that the performance of the mixture does not depend significantly on the size of the router. Furthermore, we show that our approach outperforms less complex routing methods, such as the TF-IDF encoding with K-Means clustering proposed by

Gururangan et al. (2023). Moreover, we demonstrate that the perplexity gains are not driven by a few dominant experts but result from each expert contributing positively to the overall performance.

## 3.1 EXPERIMENTAL SETUP

We use the RedPajama-V2 dataset (Computer, 2023), which is a large-scale collection of text data designed for training language models. The dataset is built from the ground up based on publicly available web data, consisting of $84$ crawls provided by Common Crawl (2024).

Both the experts and routers utilize a Transformer architecture (Vaswani, 2017) with rotary positional encoding (Su et al., 2024). All experiments use sequences of $1,024$ tokens. We train the models using the AdamW optimizer (Loshchilov & Hutter, 2017) with $\beta_1 = 0.9$, $\beta_2 = 0.99$, and a weight decay of $0.1$. Gradient clipping is applied with a maximum norm of $0.1$.

For the experts, we employ a learning rate schedule featuring a linear warm-up to $5 \times 10^{-4}$ over the first $3,000$ steps, followed by a cosine decay for the remainder of the training period. We experiment with two model sizes: a 335M-parameter model with 4, 8, 16, and 32 experts and a 1.3B-parameter model with 4, 16, and 32 experts. Tables 1 and 2 describe the architectures and the training parameters in more detail.

For the routers, we use models with 4.4M parameters. The routers are trained for $128,000$ steps using a constant learning rate of $1 \times 10^{-4}$, following a linear warm-up over the first $1,000$ steps. Scheduler choice described in details in App. A. Routers are trained with a batch size of 32 and perform all-to-all communication of the loss on the dataset chunk approximately every $45$ million training tokens (see App. A.4). The length of the prefix used for routing is set to 256 tokens, which is $25\%$ of the context length.

**Comparison to the Dense Model**  To ensure a fair comparison to the dense baseline, we designed our experiments such that the computational cost during inference and the total computational cost during training are the same. We achieve this by comparing the performance of models where each expert has the same architecture and number of parameters as the dense model, resulting in the same inference cost. In addition, we train SMALLTALK LM for the same number of total tokens which means we spend the same number of total training FLOPs.

## 3.2 LANGUAGE MODELING RESULTS

Across all experimental configurations, our method *consistently outperforms the baseline* while utilizing the same volume of data, with only minor increases in computational costs. Specifically, for the 1.3B parameter model, our approach results in less than a 1% increase in training FLOPs and less than a 3% increase in inference FLOPs, while achieving performance gains of up to almost 18% in perplexity. Similarly, for the 335M parameter model, the cost increases are modest, with less than a 4% increase in training FLOPs and less than a 10% increase in inference FLOPs, alongside performance gains of up to nearly 14% in perplexity (see Appendix A.3 for details on FLOPs calculation). Figures 2a and 2b illustrate the perplexity on a held-out test set relative to the total training cost, measured in petaFLOPs, across different model sizes. Additionally, Figure 2c shows the test perplexity as a function of the number of training tokens for the 1.3B parameter model.

Moreover, it is interesting to compare our model with the smaller 335M parameter experts to the 1.3B dense baseline. When using 32 experts and roughly the same training budget in FLOPs, SMALLTALK LM achieves slightly better perplexity (9.07 vs 9.11) but requires **three times less** compute during inference.

Finally, we observed that for a constant model size, the improvement in perplexity increases with the number of experts. This suggests that our method more efficiently utilizes the training data to improve performance, likely due to the specialization of experts on different segments of the data distribution.

**Communication Overhead**  Our approach performs minimal communication and does not require a fast interconnect. As detailed in § 2, we first train the routers which are then used to efficiently shard the dataset and each expert is trained completely independently. During their training, the routers communicate approximately 100 times with each transmission involving less than 6

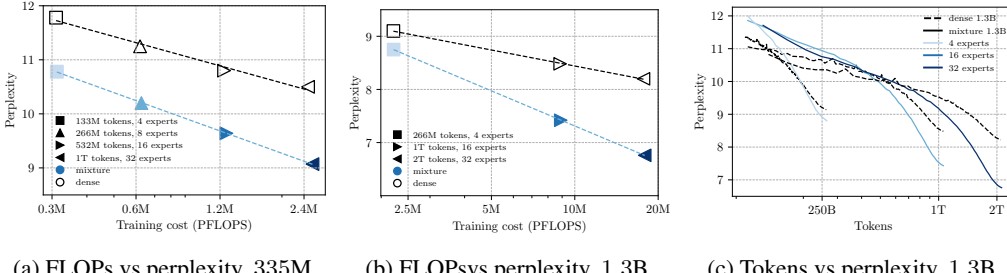

(a) FLOPs vs perplexity, 335M.     (b) FLOPsvs perplexity, 1.3B.     (c) Tokens vs perplexity, 1.3B.

Figure 2: **Better perplexity for the same price.** Test perplexity comparison between our approach and the dense baseline, as a function of training cost measured in PFLOPs. In **(a)**, we report results for models with 335M parameters using 4, 8, 16, and 32 experts and in **(b)** for models with 1.3B parameters using 4, 16, and 32 experts. In addition, **(c)** shows the perplexity comparison between our approach and the dense baseline, plotted against the cumulative number of tokens processed throughout the training for the 1.3B parameter models. We observe that our method significantly outperforms the baseline across all experimental configurations. Notably, our 335M parameter model with 32 experts achieves a perplexity of 9.07, outperforming the 1.3B dense baseline's perplexity of 9.1. This improvement is achieved with a training budget of $2.5 \times 10^{21}$ FLOPs, which is comparable to the baseline's $2.2 \times 10^{21}$ FLOPs, while requiring **three times less** computational cost during inference ($0.87 \times 10^{12}$ FLOPs compared to $2.81 \times 10^{12}$ FLOPs). See § 3.2 and App. A for a detailed description of our experimental setup.

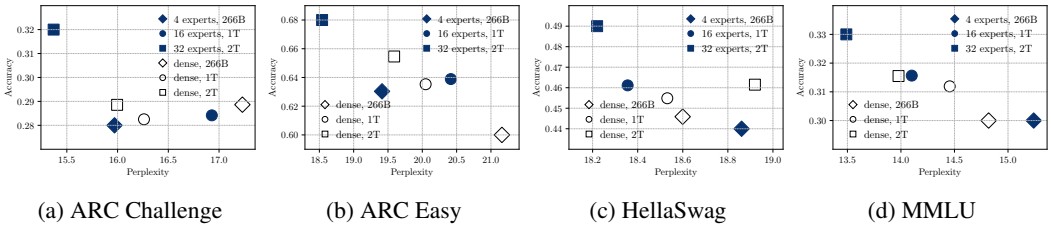

(a) ARC Challenge     (b) ARC Easy     (c) HellaSwag     (d) MMLU

Figure 3: **Downstream evaluation.** Accuracy with respect to perplexity on **(a)** ARC Challenge, **(b)** ARC Easy, **(c)** HellaSwag and **(d)** MMLU, for 1.3B parameter dense baselines trained on 266B, 1T and 2T tokens (empty symbols) and mixture models with 1.3B parameter experts and 4, 16 and 32 experts respectively (filled symbols). The models that have the same symbol shape have near identical training and inference FLOPs.

megabytes per router resulting in truly minimal communication requirements. See A.4 for details about the calculation of these numbers.

## 3.3 DOWNSTREAM EVALUATION

To demonstrate the applicability of our approach in real-world scenarios, we performed zero-shot evaluation on a set of natural language processing (NLP) tasks, including ARC Challenge and ARC Easy (Clark et al., 2018), HellaSwag (Zellers et al., 2019), SciQ (Welbl et al., 2017), and MMLU (Hendrycks et al., 2020). Figure 3 shows both accuracy and perplexity for each task (see Appendix B for details regarding perplexity and accuracy computation).

Specifically, our largest configuration, a model with 32 1.3B parameter experts, demonstrates better performance on four out of five tasks, achieving gains of 3%, 2%, 3%, and 1% on ARC Challenge (3a), ARC Easy (3b), HellaSwag (3c), and MMLU (3d), respectively, under the same inference cost.

Moreover, on the MMLU benchmark, our approach either beats the baseline or achieves the same performance on 75% of the tasks (42 out of 56), as shown in Table 4.

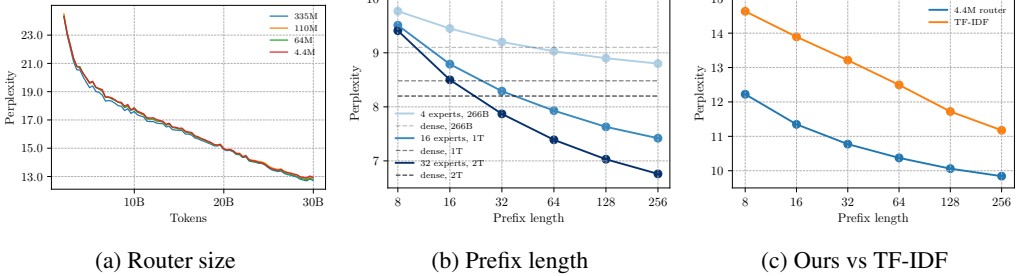

|  (a) Router size | (b) Prefix length | (c) Ours vs TF-IDF |

Figure 4: **Routing analysis. (a)** Test perplexity over training steps for different router sizes using a 335M parameter model with 4 experts. We compare routers of sizes 335M (where the model routes data for itself), 110M, 65M, and 4.4M parameters. **(b)** Test perplexity as a function of routing prefix length during inference for 1.3B parameter model with 4, 16 and 32 experts. We examine how reducing the prefix length $\hat{M}$ used during inference affects performance when the data is partitioned during training using a prefix size $M \geq \hat{M}$. **(c)** Test perplexity over training steps for a 335M parameter model with 16 experts, comparing our proposed routing using TF-IDF document encoding followed by SVD projection and balanced K-Means clustering.

## 3.4 ROUTING ANALYSIS

In this section we analyse the most critical component of SMALLTALK LM, the routing. In particular, we perform experiments to investigate the impact of the size of the router to the mixture model, the impact of the size of the prefix and finally whether the EM algorithm presented in § 2 is critical or it could be replaced by simple content based clustering.

**Impact of Router Size**    One of the most critical findings of our work is the fact that *the size of the model* used for routing *does not impact* the performance of the mixture (Figure 4a). This allows us to utilize small router models which reduce dramatically the computational cost of routing during training and inference without compromising the model's performance.

In detail, we investigated the effect of router size on the performance of our method by experimenting with a 335M parameter model configured with 4 experts, using four different router sizes for data partitioning: a 335M parameter router (where the model routes data for itself, optimizing for the data it handles best), and routers with 110M, 64M, and 4.4M parameters. All models were trained with a batch size of 32 for 256, 000 steps, using a routing prefix length of 256 tokens, following the same experimental setup as in § 3.2 (see Appendix A for more details about model architectures and training parameters). Figure 4a illustrates the test perplexity wrt the number of training tokens for these configurations. We observe that all router sizes perform practically identically.

**Impact of Prefix Length**    An important hyperparameter for our mixture model is the length of the context used for routing to the expert model. In our experiments, we use a prefix length of 256 which may be too large for some real-world applications like conversational AI. However, we show in Figure 4b that SMALLTALK LM outperforms the dense baseline even when routing with shorter prefixes. In particular, we show that even with prefixes of just 32 tokens ($\frac{1}{8}$th of the length during training) the model still outperforms the dense baseline.

**Comparison With Routing Using TF-IDF**    An alternative approach might suggest that if a tiny LLM suffices for routing, simpler methods such as TF-IDF encoding combined with balanced K-Means clustering, as proposed by Gururangan et al. (2023), could be adequate. To test this hypothesis, we trained a 335M parameter model using the routing strategy outlined in (Gururangan et al., 2023): applying TF-IDF transformation to the text, followed by Singular Value Decomposition (SVD) projection into a low-dimensional space, and then clustering using balanced K-Means.

We then trained 16 experts on these clustered results. Our findings reveal that routing with TF-IDF encoding under performs when the prefix is short, indicating limitations of this simple routing approach. As shown in Figure 4c, our proposed routing method significantly outperforms the TF-

IDF-based routing. This demonstrates that our approach is more effective in capturing the nuances of the data distribution, even when using limited context for routing decisions.

**Experts Do Specialize**    A potential concern is that the observed performance gains might be driven primarily by a few experts among the $E$, while the remaining contribute minimally or may even under perform on their assigned data. To address this concern, we conducted an analysis of each expert's performance individually. Figures 5a, 5b, and 5c compare the perplexity achieved by our method and the dense baseline on the routed dataset segments for the 1.3B parameter model, utilizing mixtures of 4, 16, and 32 experts, respectively.

Our findings demonstrate that all experts contribute positively, with consistent perplexity improvements observed across all routed dataset segments. Moreover, the performance gap between our method and the baseline widens as the number of experts increases, suggesting enhanced specialization due to finer partitioning of the data distribution.

Notably, despite the absence of capacity constraints on experts during inference, unlike the enforced capacities during training, all experts are actively utilized and receive substantial portions of the data. The consistent improvements across all routed dataset segments confirm that the experts do specialize and they all contribute to the overall improvement instead of only a few dominant experts.

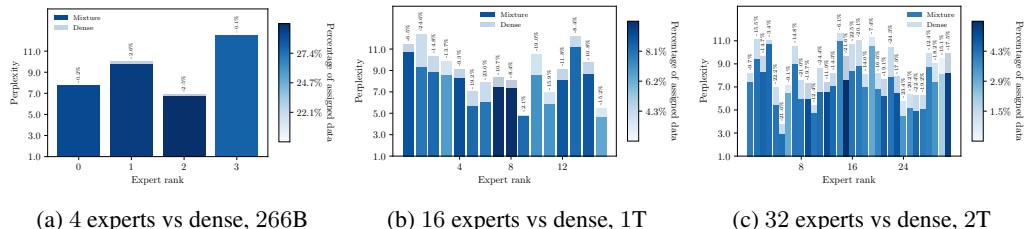

(a) 4 experts vs dense, 266B    (b) 16 experts vs dense, 1T    (c) 32 experts vs dense, 2T

Figure 5: **Experts Do specialize.** Test perplexity comparison between our method and the dense baseline on the routed dataset segments for the 1.3B parameter model, using mixtures of 4 experts in **(a)** (trained on 266B tokens), 16 experts in **(b)** (1T tokens), and 32 experts in **(c)** (2T tokens). Each bar represents a dataset segment, with the color intensity indicating the percentage of data assigned to that expert – darker shades correspond to a higher proportion of data. Overlapping bars depict the perplexity achieved by the dense baseline (translucent) and our proposed mixture model (opaque). The results demonstrate that all experts specialize effectively on their assigned segments of the data distribution, leading to consistent improvements over the baseline. While the data distribution among experts is not perfectly even – with some experts receiving more data than others – all experts receive a substantial portion of the data. This shows that each expert contributes meaningfully to the overall performance gains.

## 4    RELATED WORK

Data-parallel training requires synchronization of the gradients after each backward pass, which can be slow if the network bandwidth is limited. In this section, we discuss existing methods that aim to make data parallelism more communication-efficient via reducing the frequency of the synchronization. Since our paper also explores methods for efficient inference without performance degradation, we discuss this branch of research as well.

**Distributed optimization**    To reduce the frequency of synchronisation during training of large models, researchers have explored distributed optimization techniques both theoretically and empirically (Mcdonald et al., 2009; Stich, 2019; Lian et al., 2018; Wang & Joshi, 2019; McMahan et al., 2017; Lin et al., 2018; Zinkevich et al., 2010). The core principle of these methods is to allow each worker to execute several local training iterations before engaging in global synchronization – each device performs multiple optimizer updates per communication round. This technique also has been studied for LLM pre-training (Diskin et al., 2021; Douillard et al., 2023) and fine-tuning (Borzunov et al., 2022; Liu et al., 2024; Hilmkil et al., 2021; Ro et al., 2022).

While theses approaches reduces the frequency of communication, thereby decreasing communication overhead, they often result in degraded model performance compared to every step synchroniza-

tion (Diskin et al., 2021), or they lack direct performance comparisons in the context of pre-training (Douillard et al., 2023). The degradation in performance is often attributed to the staleness of gradients and the divergence between local and global models due to delayed synchronization (Zhang et al., 2016; Yu et al., 2019; Mitra et al., 2021). Moreover, to our knowledge, all experiments have been conducted at a small scale ($\leq$ 400M parameters) (Douillard et al., 2023; Liu et al., 2024; Ro et al., 2022). Therefore, scalable solutions that can reduce communication frequency for LLM training without compromising performance at larger scales are still needed.

**Mixture of Experts**  Mixture of Experts (MoE) models have emerged as a prominent approach to increase model capacity without a proportional increase in inference costs (Jacobs et al., 1991; Jordan & Jacobs, 1994). The key idea behind MoE is to partition the input space into subsets corresponding to specific subtasks, each managed by a specialized expert. By training different experts on distinct tasks, MoE models aim to enhance performance while reducing computational overhead during inference.

Early implementations of MoE employed algorithms such as Gaussian Processes (Tresp, 2000; Theis & Bethge, 2015; Deisenroth & Ng, 2015), Dirichlet Processes (Shahbaba & Neal, 2009), and Support Vector Machines (SVMs) (Collobert et al., 2001) to model these specialized functions. Recent advancements have integrated deep neural networks as experts, providing scalable and flexible implementations suitable for large-scale models.

In natural language processing, researchers have generalized the original MoE idea to enhance large-scale language models. For instance, Shazeer et al. (2017) introduced a Mixture of Experts feed forward layer in which, in contrast to prior works, routing decision is made for every token. This layer takes a token representation as an input and routes it to the best-matched top-$k$ experts. By doing so the number of active parameters is decreased. Similar approaches have been effectively employed in models like the Switch Transformer (Fedus et al., 2022), GShard (Lepikhin et al., 2021), GLaM (Du et al., 2022) and Mixtral of Experts (Jiang et al., 2024) which leverage MoE layers to achieve state-of-the-art results with reduced number of active parameters costs. However, since the routing decision in models like Switch Transformer MoE is made on the token level: (1) all experts must be retained in RAM during both training and inference phases, necessitating substantial memory resources, and (2) a high communication overhead is required for data transfer and gradient synchronization across experts (Lepikhin et al., 2021; Fedus et al., 2022; Riquelme et al., 2021). In that respect this approach is completely orthogonal to our proposed SMALLTALK LM. In fact, each of the experts in our method could be such a token-level mixture model which would result in even smaller inference FLOPs for a specific achieved perplexity albeit with a significantly more complex implementation and higher RAM requirements during inference.

**Independent Training of Mixture of Experts Models**  To train experts fully in parallel, recent research has also explored independent training of the mixture of experts models (Gross et al., 2017; Li et al., 2022; Gururangan et al., 2023). For instance, Gross et al. (2017) proposed to train a gater independently from the experts and use the output of the gater to hard assign the input image to experts. In language processing, Li et al. (2022) and Gururangan et al. (2023) proposed methods that train expert models separately and merge them post-training. However, these approaches rely on full document context or metadata for routing, which can be impractical in real-time applications involving shorter inputs. For instance, Gururangan et al. (2023) focus on performance in text classification tasks, which may not fully reflect real-world scenarios.

## 5 CONCLUSION

In this work, we present SMALLTALK LM – a method that significantly reduces communication overhead during LLM training. It also outperforms dense model baselines trained with a similar amount of FLOPs and the same data volume and comparable inference costs. In addition, our proposed lightweight routing with short prefixes makes SMALLTALK LM practical for real-world downstream tasks, where we show it largely outperforms the dense baselines.

These findings open up multiple directions for future research, including exploring different routing strategies for data partitioning among experts and investigating further the impact on downstream task performance. Another promising avenue is scaling the mixture to hundreds or even thousands of experts, which could further enhance model performance while benefiting from sparse inference.

ACKNOWLEDGMENTS

We thank Jagrid Digani and Awni Hannun for their insightful discussions on the algorithm and experiments. We thank Justin Deschenaux for his contributions to the early stages of the project.

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

CONTENTS

## A  LANGUAGE MODELING EXPERIMENTS

### A.1  ARCHITECTURE AND TRAINING DETAILS

Our architecture is based on the transformer decoder (Vaswani, 2017). We use rotary positional encoding (Su et al., 2024). We use a SentencePiece (Kudo & Richardson, 2018) tokenizer with a vocabulary of 32000 tokens. The model specific parameters for the different sizes are summarized in Table 1. We implemented our EM scheme for training the routers in PyTorch (Paszke et al., 2019). After segmenting the training set, the experts were trained independently using Jax (Bradbury et al., 2018). All training was done in `bfloat16`. The optimizer states and operations are in `float32`. The training parameters for specific models are summarised in Table 2.

We employ different learning rate schedules for routers and experts due to their distinct roles. Experts are trained for accurate next-token prediction, using a cosine decay learning rate to optimize their performance. In contrast, routers are used solely for routing purposes. We only need an estimation of how well a router performs on a prefix of a sequence relative to other routers. Since we are concerned with relative performance rather than absolute accuracy, applying the same training strategy to all routers is sufficient for effective routing decisions. Moreover, using a constant learn-

ing rate for routers reduces computational overhead and eliminates the need to tune the number of training steps.

| Model parameters | Role | Hidden size | FFW expansion factor | Layers | Attention heads |
|---|---|---|---|---|---|
| 335M | expert | 1024 | 4 | 24 | 16 |
| 1.3B | expert | 2048 | 4 | 24 | 16 |
| 4.4M | router | 96 | 4 | 12 | 12 |
| 64M | router | 416 | 4 | 12 | 12 |
| 110M | router | 768 | 4 | 12 | 12 |

Table 1: **Model parameters.**

| Model | Steps (k) | Tokens (B) | Batch size | # GPUs |
|---|---|---|---|---|
| 335M (dense) | 256 | 133 | 512 | 8 |
| 335M (4 experts) | 256 | 133 | 128 | 8 |
| 335M (dense) | 512 | 266 | 512 | 32 |
| 335M (8 experts) | 256 | 266 | 128 | 8 |
| 335M (dense) | 1024 | 532 | 512 | 32 |
| 335M (16 experts) | 256 | 532 | 128 | 8 |
| 1.3B (dense) | 512 | 266 | 512 | 32 |
| 1.3B (4 experts) | 512 | 266 | 128 | 8 |
| 1.3B (dense) | 1024 | 1000 | 1024 | 64 |
| 1.3B (16 experts) | 512 | 1000 | 128 | 8 |
| 1.3B (dense) | 1024 | 2000 | 2048 | 128 |
| 1.3B (32 experts) | 512 | 2000 | 128 | 8 |
| 4.4M (router) | 128 | 4 | 32 | 1 |

Table 2: **Training parameters.**

## A.2 NOTATION

For the dense baseline LLM and expert LLM:

- Number of layers ($L$)
- Hidden dimension size ($H$)
- Number of attention heads ($A$)
- Sequence length ($S$)
- Batch size ($B$)
- Vocabulary size ($V$)
- Feedforward network dimension ($D_{\text{ff}}$)

For the router LLM:

- Prefix length for routing ($M$)
- Number of layers ($L_r$)
- Hidden dimension size ($H_r$)
- Number of attention heads ($A_r$)
- Batch size ($B_r$)
- Feedforward network dimension ($D_{r_{\text{ff}}}$) – parameters specific to the router model

- Number of training tokens per router between communications $T$

We also define $N_{\text{steps}}$, $N_{\text{steps\_expert}}$, and $N_{\text{steps\_router}}$ as the number of training steps for the dense baseline, the expert in the mixture, and the router, respectively. The sequence length ($S$) and vocabulary size ($V$) are consistent across the dense baseline, router, and expert models. Finally, we denote $E$ as the number of experts (routers).

## A.3 COMPUTATIONAL COST

### A.3.1 TRANSFORMER

To estimate the computational cost of Transformer model during training and inference, we calculated the total floating-point operations (FLOPs) based on the model's architectural parameters.

For **training**, we computed the FLOPs per training step by summing the contributions from the embedding layer, multi-head attention (MHA), feedforward layers (FFN), and the output projection layer. The calculations are as follows:

- **Embedding Layer:** Although embeddings are typically implemented as lookups with negligible computational cost, for completeness, we estimate the FLOPs as:

$$\text{FLOPs}_{\text{emb}} = B \times S \times H$$

- **Multi-Head Attention (MHA):**
    1. Linear Projections (Queries, Keys, Values):
    $$\text{FLOPs}_{\text{proj}} = 3 \times 2 \times B \times S \times H \times H = 6 \times B \times S \times H^2$$

    2. Scaled Dot-Product Attention:
    $$\text{FLOPs}_{\text{attn}} = \text{FLOPs}_{\text{QK}} + \text{FLOPs}_{\text{V}} = 2 \times B \times S^2 \times H + 2 \times B \times S^2 \times H = 4 \times B \times S^2 \times H$$

    3. Output Projection:
    $$\text{FLOPs}_{\text{out\_proj}} = 2 \times B \times S \times H \times H$$

    Total Multi-Head Attention (MHA):

    $$\begin{aligned}
    \text{FLOPs}_{\text{MHA}} &= \text{FLOPs}_{\text{proj}} + \text{FLOPs}_{\text{attn}} + \text{FLOPs}_{\text{out\_proj}} \\
    &= 6 \times B \times S \times H^2 + 4 \times B \times S^2 \times H + 2 \times B \times S \times H^2 \\
    &= 8 \times B \times S \times H^2 + 4 \times B \times S^2 \times H
    \end{aligned}$$

- **Feedforward Network (FFN):**

$$\text{FLOPs}_{\text{FFN}} = 2 \times 2 \times B \times S \times H \times D_{\text{ff}} = 4 \times B \times S \times H \times D_{\text{ff}}$$

- **Output projection:**

$$\text{FLOPs}_{\text{out}} = \text{FLOPs}_{\text{out}_{\text{proj}}} + \text{FLOPs}_{\text{softmax}} = 2 \times B \times S \times H \times V + 3 \times B \times S \times V$$

**Total FLOPs per Layer:**

$$\text{FLOPs}_{\text{layer}} = (\text{FLOPs}_{\text{MHA}} + \text{FLOPs}_{\text{FFN}}) \times L$$

**Total Forward Pass FLOPs per Step:**

$$\text{FLOPs}_{\text{forward}} = \text{FLOPs}_{\text{emb}} + \text{FLOPs}_{\text{layer}} + \text{FLOPs}_{\text{out}}$$

**Total Training FLOPs per Step:**

The backward pass is approximately twice as computationally intensive as the forward pass. Therefore:

$$\text{FLOPs}_{\text{train\_step}} = \text{FLOPs}_{\text{forward}} + 2 \times \text{FLOPs}_{\text{forward}} = 3 \times \text{FLOPs}_{\text{forward}}$$

**Total Training FLOPs:**

Multiplying the FLOPs per training step by the total number of training steps:

$$\text{Total Training FLOPs} = \text{FLOPs}_{\text{train\_step}} \times N_{\text{steps}} = 3 \times \text{FLOPs}_{\text{forward}} \times N_{\text{steps}}$$

**Total Training FLOPs:**

$$
\begin{aligned}
\text{Total Training FLOPs} = 3 \times N_{\text{steps}} \times \Big[ & B \times S \times H \\
& + L \times \left( 8 \times B \times S \times H^2 + 4 \times B \times S^2 \times H + 4 \times B \times S \times H \times D_{\text{ff}} \right) \\
& + 2 \times B \times S \times H \times V + 3 \times B \times S \times V \Big]
\end{aligned}
\tag{10}
$$

**Total Inference FLOPs:**

$$
\begin{aligned}
\text{Total Inference FLOPs} = \Big[ & B \times S \times H \\
& + L \times \left( 8 \times S \times H^2 + 4 \times S^2 \times H + 4 \times S \times H \times D_{\text{ff}} \right) \\
& + 2 \times S \times H \times V + 3 \times S \times V \Big]
\end{aligned}
\tag{11}
$$

### A.3.2 MIXTURE OF LLMS

The total training FLOPs for the mixture model consist of four main components: training routers, shading data for routers, training experts, and shading data for the experts.

For a model trained with batch size $B$ over $N_{\text{steps}}$ steps, the total number of sequences shaded can be estimated as $N_{\text{steps}} \times B \times E$, where each model receives approximately $N_{\text{steps}} \times B$ sequences during training.

Thus, the FLOPs required for shading can be calculated as the inference FLOPs with an effective batch size of $N_{\text{steps}} \times B \times E$ (see Equation 11).

Therefore **Total Training FLOPs:**

$$\text{Total Training FLOPs} = \text{Training FLOPs Routers} + \text{Training FLOPs Experts} = \quad (12)$$

$$
+ 3 \times N_{\text{steps\_router}} \times \Big[ B_r \times S \times H_r
$$

$$
+ L_r \times \left( 8 \times B_r \times S \times H_r^2 + 4 \times B_r \times S^2 \times H_r + 4 \times B_r \times S \times H_r \times D_{r_{\text{ff}}} \right)
$$

$$
+ 2 \times B_r \times S \times H_r \times V + 3 \times B_r \times S \times V \Big] \times E \quad \text{(Eq. 13: Training Routers)}
$$

$$(13)$$

$$
+ \left( N_{\text{steps\_router}} \times B_r \times E \right) \times \Big[ M \times H_r
$$

$$
+ L_r \times \left( 8 \times M \times H_r^2 + 4 \times M^2 \times H_r + 4 \times M \times H_r \times D_{r_{\text{ff}}} \right)
$$

$$
+ 2 \times M \times H_r \times V + 3 \times M \times V \Big] \times E \quad \text{(Eq. 14: Shading data for routers)}
$$

$$(14)$$

$$
3 \times N_{\text{steps\_expert}} \times \Big[ B \times S \times H
$$

$$
+ L \times \left( 8 \times B \times S \times H^2 + 4 \times B \times S^2 \times H + 4 \times B \times S \times H \times D_{\text{ff}} \right)
$$

$$
+ 2 \times B \times S \times H \times V + 3 \times B \times S \times V \Big] \times E \quad \text{(Eq. 15: Training Experts)}
$$

$$(15)$$

$$
+ \left( N_{\text{steps\_expert}} \times B \times E \right) \times \Big[ M \times H_r
$$

$$
+ L_r \times \left( 8 \times M \times H_r^2 + 4 \times M^2 \times H_r + 4 \times M \times H_r \times D_{r_{\text{ff}}} \right)
$$

$$
+ 2 \times M \times H_r \times V + 3 \times M \times V \Big] \times E \quad \text{(Eq. 16: Shading data for experts)}
$$

$$(16)$$

In the above formula:

- Equations 15 and 13 correspond to the total training FLOPs for $E$ experts and $E$ routers respectively.
- Equations 16 and 14 represents the inference cost of $E$ routers on $B$ batches with a prefix length $M$ used to partition the data for experts and routers respectively.

Table 3 presents the training and inference costs for our proposed mixture of language models compared to the dense baseline LLMs.

The expert balancing during training introduces only a negligible computational overhead relative to both training and inference costs. Therefore, it is excluded from our FLOPs calculations.

| Model | Perplexity | Training cost | Inference cost |
|---|---|---|---|
| 335M (dense) | 11.78 | 31.02 | 0.79 |
| 335M (4 experts) | $10.78 \downarrow 8.49\%$ | $31.02 + 0.22 \uparrow 0.07\%$ | $0.79 + 0.01 \uparrow 1.27\%$ |
| 335M (dense) | 11.25 | 62.03 | 0.79 |
| 335M (8 experts) | $10.20 \downarrow 9.33\%$ | $62.03 + 0.75 \uparrow 1.20\%$ | $0.79 + 0.02 \uparrow 2.53\%$ |
| 335M (dense) | 10.80 | 124.06 | 0.79 |
| 335M (16 experts) | $9.64 \downarrow 10.74\%$ | $124.06 + 2.71 \uparrow 2.19\%$ | $0.79 + 0.04 \uparrow 5.06\%$ |
| 335M (dense) | 10.5 | 248.12 | 0.79 |
| 335M (32 experts) | $9.07 \downarrow 13.62\%$ | $248.12 + 10.28 \uparrow 4.14\%$ | $0.79 + 0.08 \uparrow 10.13\%$ |
| 1.3B (dense) | 9.10 | 221.33 | 2.81 |
| 1.3B (4 experts) | $8.75 \downarrow 3.85\%$ | $221.33 + 0.36 \uparrow 0.16\%$ | $2.81 + 0.01 \uparrow 0.36\%$ |
| 1.3B (dense) | 8.48 | 885.32 | 2.81 |
| 1.3B (16 experts) | $7.42 \downarrow 12.53\%$ | $885.32 + 4.87 \uparrow 0.55\%$ | $2.81 + 0.04 \uparrow 1.42\%$ |
| 1.3B (dense) | 8.20 | 1770.65 | 2.81 |
| 1.3B (32 experts) | $6.76 \downarrow 17.56\%$ | $1770.65 + 18.94 \uparrow 1.07\%$ | $2.81 + 0.08 \uparrow 2.85\%$ |

Table 3: **Performance gain versus computational overhead.** The table compares performance improvements and computational costs for models with 335M (trained with 4, 8, 16, and 32 experts) and 1.3B parameters (trained with 4, 16, and 32) experts. Training cost is measured in $10^{19}$ FLOPs, inference cost is measured in $10^{12}$ FLOPs. The dense baselines and corresponding mixture models were trained on the same data volume.

**Regarding Computational Overhead** As demonstrated in Table 3, the 1.3B parameter model with 32 experts achieves significant performance improvements almost cost-free, gaining nearly 18% in perplexity compared to the dense baseline while incurring only 1% additional cost during training and less than 3% during inference. Furthermore, a 335M parameter mixture of 32 experts reaches the same performance level as the 1.3B parameter model, but with a comparable training budget and **three times lower inference cost**. Despite the 335M parameter model incurring a higher percentage cost during inference, this computational overhead is not the definitive minimum necessary for effective routing: as detailed in § 3.4, our results indicate that the routing overhead can be significantly reduced by utilizing smaller routers and shorter prefix sizes. This suggests that further reductions in computational costs, without compromising routing quality, are feasible and present a valuable direction for future research.

## A.4 COMMUNICATION OVERHEAD

In this section, we provide an estimate of the communication overhead involved in our model training compared to a dense baseline trained with standard distributed data parallelism. We quantify both the volume of data transmitted and the frequency of communication required during training. For simplicity of computation, we assume that all collective communications are bandwidth-optimal, meaning that the total number of bytes transferred per node is approximately $2K$, where $K$ is the size of the message (Thakur et al., 2005; Patarasuk & Yuan, 2009).

**Mixture of LLMs** Every router performs all gather operation to share the loss on the dataset chunk approximately every $T = 45M$ training tokens, as outlined in 3.2. This frequency was primarily determined by implementation specifics related to data storage.

We can define the number of tokens processed per step by a router as $N_{\text{tokens\_per\_step}}$, which can be calculated as:

$$N_{\text{tokens\_per\_step}} \leq S \times B_r$$

This leads to the minimum number of steps between communications, $N_{\text{steps\_per\_com}}$, given by:

$$N_{\text{steps\_per\_com}} \geq \frac{T}{S \times B_r}$$

The maximum number of communication events, $N_{\text{comm}}$, during the router's training can then be estimated as:

$$N_{\text{comm}} \leq \frac{N_{\text{steps\_router}}}{N_{\text{steps\_per\_comm}}} = \frac{N_{\text{steps\_router}} \times S \times B_r}{T}$$

As shown in App. Table 2 we train routers for $128,000$ steps with batch size 32. We use context size 1024. Therefore, $N_{\text{comm}} \approx 94 < 1 \times 10^2$.

The total amount of data sent and received by each node (router) can be calculated as the total number of sequences multiplied by the number of bytes per sequence. We consider that the loss is transferred in `float16`, which requires 2 bytes per element. The total number of sequences can be estimated as $\frac{T \times N}{S}$.

Each router needs to send and receive the loss values for each sequence to and from other routers, resulting in a factor of 2 for the data sent and received. Therefore, the data per router can be approximated as:

$$\text{Data per router} \approx 2 \times 2 \times \frac{T \times E}{S}$$

Given $T = 45 \times 10^6$ – training tokens per router between communication steps, $E \leq 32$ – number of experts and $S = 1,024$ – sequence length, we can estimated the data transferred per node as:

$$\text{Data per router} \leq 2 \times 2 \times \frac{T \times E}{S} = 4 \times \frac{45 \times 10^6 \times 32}{1,024} = 5.625\text{MB}$$

Once routers are fully trained, they communicate only when necessary, such as when experts require new data for training.

For instance, consider that the message size is $K$ bytes, equivalent to loss array of $K/2$ sequences in `float16` precision. The frequency of communication after routers are fully trained is determined by the batch size of the expert $B$ and the number of experts $E$. Specifically, the number of steps each expert makes between communications can be estimated as:

$$N_{\text{steps\_per\_comm}} \approx \frac{K}{2 \times B \times E}$$

Thus, communication can be scheduled approximately every $\frac{K}{4 \times B \times E}$ expert's training steps. Assuming $K = 2^{31} - 1$ bytes, $B = 128$ as used in the majority of our experiments, and $E \in \{4, 8, 16, 32\}$ (see Appendix 2), the communication interval becomes:

$$\frac{2^{31} - 1}{2 \times 128 \times E} \approx \frac{2^{23}}{E} \leq 2^{18} \quad \forall E \in \{4, 8, 16, 32\} \tag{17}$$

**Comparison with Distributed Training**   Let us assume that we train a model with a batch size of $B$ on a single host, where $W$ represents the model size in parameters. For simplicity, we consider only data parallelism and assume that the model weights can fit on one GPU. To compare communication costs, we examine the additional expense incurred when increasing throughput by a factor of $E$, i.e., utilizing $E$ hosts instead of one.

In regular distributed training, scaling to $E$ hosts necessitates copying all weights to each host and performing gradient synchronization at every training step. Assuming each parameter is represented by 32 bits (as optimizers typically operate in `float32`), the amount of data transferred to and from each node during gradient synchronization can be calculated using the bandwidth-optimal algorithm:

$$\text{Data per node} = 2 \times W \times 4$$

Here, $W \times 4$ represent the size of the model gradients in bytes.

If $W = 1.3 \times 10^9$ parameters, then the amount of data transferred per node per training step is:

$$\text{Data per node} = 2 \times 1.3 \times 10^9 \times 4 = 10.4\text{GB}$$

Therefore, in regular distributed training, each node must send and receive **gigabytes** of data **at every step**. In contrast, our approach offers flexible communication strategies. We can transfer large volumes of data only a few times during training (see Equation 17), or optimise for more frequent communications with smaller data sizes. This flexibility allows us to adapt based on network latency and other infrastructure constraints.

## B  DOWNSTREAM EVALUATION

For the evaluation, we utilized the *lm-eval-harness* software (Gao et al., 2024). For the computation of perplexity, we compute the perplexity using the format: "Question: {question}. Answer: {answer}."

| Task | 32 experts, 2T | dense, 2T |
|---|---|---|
| mmlu abstract algebra | 0.1400 | **0.2300** |
| mmlu astronomy | **0.3553** | 0.2829 |
| mmlu business ethics | 0.4900 | **0.5300** |
| mmlu college computer science | 0.2500 | **0.2900** |
| mmlu college mathematics | **0.1500** | 0.1400 |
| mmlu computer security | 0.3500 | **0.3800** |
| mmlu elementary mathematics | **0.2672** | 0.2487 |
| mmlu formal logic | **0.3016** | 0.2937 |
| mmlu global facts | **0.3100** | 0.2800 |
| mmlu high school geography | **0.3586** | **0.3586** |
| mmlu high school macroeconomics | **0.3128** | 0.2872 |
| mmlu high school microeconomics | **0.3193** | **0.3193** |
| mmlu high school statistics | **0.2917** | 0.2593 |
| mmlu human aging | **0.4081** | 0.3857 |
| mmlu human sexuality | **0.3969** | 0.3893 |
| mmlu international law | **0.2314** | **0.2314** |
| mmlu logical fallacies | **0.3067** | 0.2945 |
| mmlu machine learning | 0.2589 | **0.2679** |
| mmlu medical genetics | **0.3900** | 0.3500 |
| mmlu moral disputes | 0.2803 | **0.2890** |
| mmlu moral scenarios | **0.2380** | **0.2380** |
| mmlu nutrition | **0.3039** | 0.2516 |
| mmlu prehistory | **0.4167** | 0.3858 |
| mmlu professional accounting | **0.2872** | 0.2518 |
| mmlu professional medicine | **0.3566** | 0.2904 |
| mmlu professional psychology | **0.3203** | 0.3186 |
| mmlu world religions | **0.5731** | 0.4503 |
| mmlu anatomy | **0.4667** | 0.3926 |
| mmlu clinical knowledge | **0.3434** | 0.2755 |
| mmlu college biology | **0.4028** | 0.3819 |
| mmlu college chemistry | **0.2800** | 0.2600 |
| mmlu college medicine | **0.2832** | 0.2659 |
| mmlu college physics | 0.1863 | **0.1961** |
| mmlu conceptual physics | **0.3787** | **0.3787** |

Table 4: **Downstream Evaluation Results (Part 1).** This table presents a comparison in accuracy on downstream tasks between a 1.3B-parameter, 32-expert model and a 1.3B-parameter dense model. Both models were trained using the 2T tokens and nearly the same training FLOPs, with only a $1\%$ difference. The inference cost of the expert model increased by less than $3\%$ compared to the dense baseline FLOPs.

| Task | 32 experts, 2T | dense, 2T |
|---|---|---|
| mmlu econometrics | **0.2719** | 0.2018 |
| mmlu electrical engineering | 0.2552 | **0.2621** |
| mmlu high school biology | **0.3290** | 0.3097 |
| mmlu high school chemistry | **0.2167** | 0.1970 |
| mmlu high school computer science | **0.2400** | **0.2400** |
| mmlu high school european history | 0.2848 | **0.3212** |
| mmlu high school government and politics | **0.4093** | 0.3834 |
| mmlu high school mathematics | **0.1519** | 0.1444 |
| mmlu high school physics | **0.2980** | 0.2715 |
| mmlu high school psychology | **0.4642** | 0.4330 |
| mmlu high school us history | **0.3382** | 0.3137 |
| mmlu high school world history | 0.2827 | **0.2996** |
| mmlu jurisprudence | **0.2407** | 0.2222 |
| mmlu management | 0.3786 | **0.3981** |
| mmlu marketing | **0.4915** | **0.4915** |
| mmlu miscellaneous | **0.5121** | 0.4994 |
| mmlu philosophy | 0.2958 | **0.3119** |
| mmlu professional law | **0.2634** | 0.2536 |
| mmlu public relations | 0.3545 | **0.4273** |
| mmlu security studies | **0.3224** | 0.3020 |
| mmlu sociology | **0.2786** | **0.2786** |
| mmlu us foreign policy | 0.3300 | **0.3400** |
| arc challenge | **0.3200** | 0.2892 |
| arc easy | **0.6780** | 0.6549 |
| hellaswag | **0.4912** | 0.4663 |
| sciq | 0.8950 | **0.9130** |

Table 5: **Downstream Evaluation Results (Part 2)**

## C ADDITIONAL RESULTS

**Effect of Prefix Length on Routing Performance**    We also conducted experiments by partitioning the data using a shorter prefix length of $M = 32$ tokens during training, whereas in our main results (see § 3), we used a prefix length of $M = 256$ tokens.

We observed that training with a smaller prefix length resulted in better expert performance when routing during inference was done using a small number of tokens (see Figure 6). This suggests that using a shorter prefix during training enhances the router's ability to make effective decisions with limited context.

We did not take any action to make sure that the router models could extrapolate to longer sequences than seen in training (e.g. interpolating positional encodings, using ALiBi (Press et al., 2022), etc.). As expected, we observe a performance degradation when making routing decisions for sequences longer than the ones used for training.

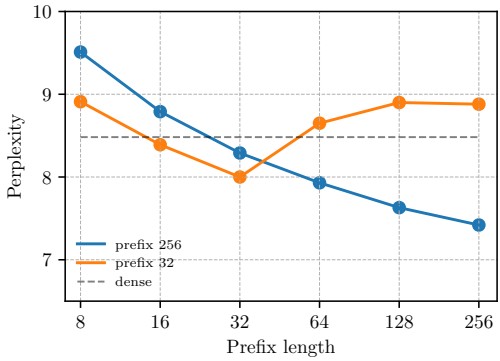

Figure 6: Test perplexity as a function of routing prefix length during inference for 1.3B parameter models with 16 experts. The orange curve represents a prefix length of 32, while the blue curve corresponds to a prefix length of 256.

