# OpenReview forum: "No Need to Talk: Asynchronous Mixture of Language Models"
_ICLR.cc/2025/Conference — ICLR 2025 Spotlight_

### Official Review · Reviewer_GV45 · 2024-11-03

**Soundness:** 3
**Presentation:** 3
**Contribution:** 2
**Rating:** 6
**Confidence:** 3

**Summary:**

This paper introduces SmalltalkLM, where instead of having a single large dense connected network, we have multiple individual dense models (the experts) and a router, which predicts which expert a given input should be routed to. During training, the data is partitioned into separate groups using the router, and each expert is then trained with language modelling on the allocated subset of data. During inference, the router can assign a new sample to the relevant expert. This enables the model to be trained with more limited RAM requirements (as each expert can be trained independently on the partitioned data) and yields better performance while only having a marginal increase in cost at inference (with the increase only as the router is first used to route the current input).

**Strengths:**

- Unlike approaches like MoE, the experts here are independent, and therefore, the routing can be done before each training epoch. Since each system can be trained independently, this reduces the simultaneous RAM requirements, which, unlike MoE approaches, enables SmalltalkLM systems to be trained using the same computational hardware. Inference also has lower RAM requirements.

- They demonstrate that this approach results in better performance across different model sizes and can yield improved downstream performance on different tasks beyond language modelling.

- The paper is quite easy to follow, and the overall idea is quite simple and seems reasonable.

**Weaknesses:**

- As far as I understand, this work takes ideas of existing deep learning MoE but simplifies the process, and instead of using the top-k experts, operating at the token-level, or having experts per layer, have a single router to a single expert, which may be of limited novelty.

- The savings in RAM might not be too substantial, as when deployed in live systems, you’ll need to load up all experts simultaneously anyway (to avoid the massive latency of continually reloading the next expert). Since the running inference costs are already more expensive than training, it seems to be quite a contrived advantage.

- this point is a bit pedantic, but the base system was presented as a vanilla LLM (i.e. not actually helpful for any particular task), although it was demonstrated it yielded marginally better performance of MCQA tasks in an ‘emergent’ setting. If these systems are further tuned (which I assume is the purpose of having these base LLMs), the routing aspect may limit the practicality. The router balancing may no longer be applicable, and possibly, specialized systems may not generalize as well as a single network trained on all the data when adapted to particular tasks. It might be interesting to see how this model might operate in more practical settings, i.e. with instruction-tuning or fine-tuning to particular tasks.

**Questions:**

- Just to clarify, the performance shown in Figure 3 is the performance of the model, which was only trained using language modelling, when it is further prompted for the downstream tasks?

- I want to raise the weaknesses above in case the authors have any further comments.

---

> ### Author Response · Authors · 2024-11-21
>
> Dear Reviewer,
>
> Thank you for your feedback, we appreciate the time that you spent reading our work and finding the idea easy to follow, simple and intuitive.
>
> We address your comments and questions below:
>
> **Comment**:  As far as I understand, this work takes ideas of existing deep learning MoE but simplifies the process, and instead of using the top-k experts, operating at the token-level, or having experts per layer, have a single router to a single expert, which may be of limited novelty.
>
> **Response**:
>
> If the reviewer believes our work lacks novelty, we would respectfully request to point out prior approaches that employ a similar prefix routing strategy and route on a sequence level using the same computational cost while achieving comparable performance in both perplexity and downstream tasks. This feedback would be valuable in helping us refine and position our contributions more clearly.
>
> **Comment**:
> The savings in RAM might not be too substantial, as when deployed in live systems, you’ll need to load up all experts simultaneously anyway (to avoid the massive latency of continually reloading the next expert). Since the running inference costs are already more expensive than training, it seems to be quite a contrived advantage.
>
> **Response**:
>
> We respectfully disagree with this comment. In our approach, because routing occurs at the sequence level, each expert processes entire sequences independently. This means we do not need to load all experts into RAM simultaneously on a single machine (compared to MoE architectures like Switch Transformers). Instead, experts can be distributed across multiple servers or devices. This eliminates the need for significant RAM on any single device.
>
> **Comment**:
> this point is a bit pedantic, but the base system was presented as a vanilla LLM (i.e. not actually helpful for any particular task), although it was demonstrated it yielded marginally better performance of MCQA tasks in an ‘emergent’ setting. If these systems are further tuned (which I assume is the purpose of having these base LLMs), the routing aspect may limit the practicality. The router balancing may no longer be applicable, and possibly, specialized systems may not generalize as well as a single network trained on all the data when adapted to particular tasks. It might be interesting to see how this model might operate in more practical settings, i.e. with instruction-tuning or fine-tuning to particular tasks.
>
> **Response**:
>
> Thank you for your comment regarding instruction tuning. We agree that exploring instruction tuning and supervised fine-tuning would be an interesting direction for future research. However, as highlighted in our contributions, the focus of this work was on pretraining and evaluating the base system without tuning it for specific tasks. We appreciate your insight and believe that extending this approach to more practical settings, including task-specific fine-tuning, is an important potential area for future exploration that we are considering for future directions.
>
> **Question**:
> Just to clarify, the performance shown in Figure 3 is the performance of the model, which was only trained using language modelling, when it is further prompted for the downstream tasks?
>
> **Response**:
>
> Thank you for your question. Yes, that is correct. We appreciate your attention to this detail and are happy to provide further clarification if needed.

---

> ### Comment · Reviewer_GV45 · 2024-11-24
> **Response to Authors**
>
> Thank you for the rebuttal. I agree that the approach has not been explored by other works (as far as I'm aware), though the approach is quite simple, though it could be argued that this simplicity might also be an advantage of the method.
>
> Also, I agree with point 2, and I hadn't considered that if you have the weights on separate machines, the maximum RAM for any single device may be reduced, which can increase the practicality of this approach.
>
> Based on the rebuttal, I have increased the score to 6.

---

> ### Author Response · Authors · 2024-11-24
>
> Thank you for carefully considering our rebuttal and for your thoughtful decision to increase your score.

---

### Official Review · Reviewer_8sbF · 2024-11-04

**Soundness:** 4
**Presentation:** 4
**Contribution:** 4
**Rating:** 8
**Confidence:** 3

**Summary:**

The study introduces a method for asynchronously training a mixture of language models (LLMs), called experts, which significantly reduces communication overhead by eliminating the need for high-volume interactions between experts during training. Each expert is specialized to process a distinct segment of data, directed by a small, specialized router—a lightweight language model with as few as 4 million parameters. These routers evaluate text sequences using a scoring system that determines the most suitable expert for each sequence based on a short prefix of 256 tokens. A novel balanced assignment approach is proposed to ensure that each expert receives a roughly equal volume of data, minimizing training imbalances. The only communication overhead is needed for the data partitioning phase and the conventional high-volume communication between experts is completely removed by the proposed method.

**Strengths:**

- The authors show that the same perplexity can be achieved with three times less compute cost compared to the dense baseline

- The final performance (test PPL) is not sensitive to the router size hence, making the approach succeed with even tiny routers (4M parameters)

- The authors have conducted experiments that are highly expensive in nature. The community will benefit from such a study. Especially, the test PPL benefits (line 349) with three times less compute is a remarkable result, it can potentially open avenues for more research on large-scale LLM development with limited resources.

**Weaknesses:**

Performance (test PPL) is sensitive to the prefix length (256 tokens used to train routers) and the sensitivity increases with the number of experts.

**Questions:**

Minor edits:
- Line 300: glitch in the reference for Adam Optimizer
- Figure 3 is difficult to read, especially the legends


Questions:
- Line 309: a different schedule (warmup + stable) for the routers, it’d be great if authors could add a line of reasoning behind this decision
- Line 388: Figure 4 (b): If the performance is sensitive to the prefix length, how would it affect the tasks where only a few tokens are provided as the prompt e.g., essay writing?

---

> ### Author Response · Authors · 2024-11-21
>
> Dear Reviewer,
>
> Thank you for your thoughtful and detailed review. We greatly appreciate your recognition that this research direction can benefit the community. Your constructive feedback is very valuable, and we are happy to address the weaknesses, questions, and minor edits you have highlighted.
>
> **Comment**: Performance (test PPL) is sensitive to the prefix length (256 tokens used to train routers) and the sensitivity increases with the number of experts.
>
> **Response**:
>
> We acknowledge that the performance sensitivity to prefix length is an important consideration. As the number of experts in the mixture increases, the routing task becomes more challenging due to the increased granularity of specialisation. This makes the model more sensitive to the length of the prefix used for routing. Intuitively, the chance of randomly assigning a sequence to the correct expert decreases linearly with the number of experts.
>
> To mitigate this sensitivity, one potential improvement is to use variable prefix lengths during training. By training the routers with a range of prefix lengths—including shorter ones—we can enhance their robustness and ability to make accurate routing decisions. We are considering this strategy for future work and thank the reviewer for this valuable insight.
>
>
> **Comment**:
> Line 309: a different schedule (warmup + stable) for the routers, it’d be great if authors could add a line of reasoning behind this decision
>
> **Response**:
>
> We chose a different learning rate schedule—a warmup phase followed by a stable constant learning rate—for the routers due to the distinct roles of routers and experts in our model. While experts are trained for accurate next-token prediction and benefit from a cosine decay learning rate to fine-tune their performance over time, routers serve a different purpose. Routers are designed to quickly learn how to assign input sequences to the appropriate experts based on a short prefix, without the need to model token distributions accurately.
> Our primary concern with the routers is their relative performance in assigning sequences compared to other routers, rather than their absolute performance in language modeling. Therefore, using the same training strategy with a constant learning rate for all routers is sufficient for effective routing decisions. This approach simplifies the training process and reduces computational overhead by eliminating the need for extensive hyperparameter tuning associated with learning rate decay schedules.
>
> Moreover, as demonstrated in Figure 4(a) of our paper, a small router with 4.4 million parameters achieves comparable routing effectiveness to a larger router with 335 million parameters trained with a cosine decay schedule.
>
> We will add the explanation to the revised version (Section 3.1 EXPERIMENTAL SETUP ) to clarify our reasoning behind using a constant learning rate. Also we will add an extended paragraph to the Appendix A.1 to elaborate on the scheduler choice.
>
> **Minor**: Line 300: Glitch in the reference for Adam Optimizer.
>
> **Response**:
>
> Thank you for bringing this to our attention. We will correct the citation in the revised manuscript.
>
> **Minor**: Figure 3 is difficult to read, especially the legends.
>
> **Response**:
>
> We appreciate your feedback regarding Figure 3 and apologize for any difficulty in reading it. In the revised manuscript, we will improve the figure by enlarging the font size of the legends and correcting the labels for the x-axis.
>
> **Comment**:
> Line 388: Figure 4 (b): If the performance is sensitive to the prefix length, how would it affect the tasks where only a few tokens are provided as the prompt e.g., essay writing?
>
> **Response**:
>
> Thank you for raising this important question about how our model performs with very short prompts.
> Our experiments indicate that despite the sensitivity to prefix length, our model remains effective with shorter prompts:
>
> ### ARC Dataset:
>
>  In the ARC dataset introduced by Clark et al. [1], the average question length is approximately 19–23 tokens (see Table 3: Properties of the ARC Dataset), which is significantly shorter than the 256-token prefix length used during our routing. However, as shown in Figure 3 of our paper, our approach archives comparable or better results on ARC tasks despite the shorter prompts.
>
> ### MMLU and HellaSwag Datasets:
>
> While datasets like MMLU [2] and HellaSwag [3] have on average longer prompts compared to ARC, a substantial portion are shorter than 128 tokens. This is depicted in Appendix B.1, Figure 11 for MMLU and Section 4.3, Figure 7 for HellaSwag.
>
> [1] https://arxiv.org/pdf/1803.05457
>
> [2] https://arxiv.org/pdf/2009.03300
>
> [3] https://arxiv.org/pdf/1905.07830

---

### Official Review · Reviewer_iShb · 2024-11-10

**Soundness:** 3
**Presentation:** 4
**Contribution:** 3
**Rating:** 8
**Confidence:** 4

**Summary:**

Large language models at modern scale are typically trained by means of distributed, synchronous model-parallel algorithms that rely on bespoke compute clusters with expensive fast interconnect hardware. To make progress towards eliminating this requirement, the authors tap into recent literature in parameter-sparse language models. So-called "mixture of experts" (MoE) language models replace a monolithic language model with a number of smaller "expert" models that are allocated inputs according to some (typically co-trained) routing mechanism and therefore require many fewer active parameters per input at inference time. The authors propose efficient new routers that can be trained completely independently of the mixture of experts, permitting asynchronous training for each expert. They show that mixture-of-expert models trained with these routers are competitive with dense language models of the same size and also outperform MoE models trained with even more trivial routers previously explored in the literature.

**Strengths:**

The method is simple (to its credit), intuitive, and effective, and it attacks an important problem. While information about the largest and most significant language models is, of course, sparse, they do not appear to be growing at the same rate as they have in previous years, and significant attention is now being devoted to the problem of improving the performance of comparatively small models (e.g. 2B models like Gemma). Good algorithms for asynchronous training could potentially unlock previously unattainable advantages of scale.

The paper is thorough and explicitly addresses most of the potential shortcomings of the proposed method that came to my mind as I read it (e.g. efficient allocation of inputs, the question of how each of the experts perform relative to each other). The authors ran a lot of very expensive experiments. It's also quite clearly written.

I was also surprised by the effectiveness of small routing models. I wouldn't have guessed that the data partitions would have been so similar across model scales.

**Weaknesses:**

I'm not in love with the choice to benchmark the asynchronous MoE against a dense model with the same number of parameters as each expert but trained for as many tokens as all experts combined (see the para. starting "Comparison to the Dense Model..."). Depending on how the number of training tokens was chosen, it might unfairly bias the results towards the expert setup---training one really saturated model up to 32x longer than each expert might just be a waste of compute and not a realistic counterfactual. At inference time, it's also a little generous to the MoE approach. It implicitly assumes that all of the experts are loaded in GPU memory at inference time. While this is not completely unrealistic, especially if we're modeling the very plausible scenario where we're running some kind of chat service and the number of incoming requests is high enough that we can pre-load all experts and still keep them saturated, it ignores the very real strength of the dense model that it can more efficiently be run in situations where these assumptions do not hold. I generally prefer Gururangan et al., 2023's approach to evaluation, which pitches the MoE against a single larger but train-compute-matched dense model. Inference-time compute is matched by ensembling the predictions of the top k experts. I like it for the additional reason that it gestures toward the scenario where the dense model is so large that it does not fit on a single compute node and requires model-parallel training (whereas individual experts don't). If this isn't the case, we can always still train the dense model without fast interconnect hardware if we're just willing to train it for a longer time, which, relatively speaking, diminishes the argument for a sparse model. There are lines in this paper that hint at something similar by comparing 335M experts to the 1.3B dense model, so I assume it wouldn't require too much work/compute to add more evaluations with that flavor.

Throughout, the paper also needs to do a better job positioning itself relative to Gururangan et al., 2023. The abstract is pretty representative; with the exception of the phrase "according to a short prefix" and the name of the method, the abstract of this paper could have been that for the prior paper, which differs from this one only in the choice of routing function. It needs to be clearer that this isn't the first paper to attempt asynchronous training using MoEs and that all the novelty lies in the routing.

**Questions:**

How were the number of tokens chosen for each configuration (see Weaknesses section above)?

---

> ### Author Response · Authors · 2024-11-22
>
> Dear Reviewer,
>
> We thank you for your valuable feedback. We are happy to hear that you found our method “simple (to its credit), intuitive, and effective”. All your comments and suggestions, especially regarding our evaluation strategy, would make our paper stronger. We really appreciate that you spent time on reading our work in depth.
>
> We address your comments below:
>
> **Comment**:
> Depending on how the number of training tokens was chosen, it might unfairly bias the results towards the expert setup---training one really saturated model up to 32x longer than each expert might just be a waste of compute and not a realistic counterfactual.
>
>
> **Response**:
>
> You raise an excellent point about the potential bias introduced by training the dense model on cumulatively the same number of tokens as experts. While we agree that this setup might unfairly favour the expert approach due to diminishing returns from extended training, our rationale for training the dense model longer was to ensure that the inference cost remained the same between the dense model and the experts setup. We aimed to demonstrate that, under the same inference budget, simply training a dense model for a longer time does not yield the same improvements in perplexity and downstream performance as scaling the number of experts does.
>
> To address your concern, we would like to highlight that we have included a comparison in our paper that aligns with your suggestion. In Appendix Table 3, we compare our expert setup with a larger dense model trained using approximately the same amount of computational resources (measured in FLOPs). Specifically, we have:
>
> | Model                | Perplexity | Training cost | Inference cost | Training Tokens (B) |
> |----------------------|------------|-------------------------|-------------------------------|---------------------|
> | **335M (32 experts)** | **9.07**   | 258.40                  | 0.87                          | 532                 |
> | **1.3B (dense)**     | 9.10       | 221.33                 | 2.81                          | 266                 |
>
> Training cost is measured in $10^{19}$ FLOPs, inference cost is measured in $10^{12}$ FLOPs.
>
> **Comment**:
> At inference time, it's also a little generous to the MoE approach. It implicitly assumes that all of the experts are loaded in GPU memory at inference time. While this is not completely unrealistic, especially if we're modelling the very plausible scenario where we're running some kind of chat service and the number of incoming requests is high enough that we can pre-load all experts and still keep them saturated, it ignores the very real strength of the dense model that it can more efficiently be run in situations where these assumptions do not hold.
>
> **Response**:
>
> Indeed, we acknowledge that loading all experts into GPU memory at inference time may not always be practical. However, we consider two main scenarios:
>
> 1) In high-throughput environments like chat services, it's feasible to distribute experts across multiple GPUs or workers, each handling a subset of experts. This allows for efficient utilisation of resources without overloading a single GPU.
>
> 2) For individual use on devices with limited GPU memory, loading experts from fast storage solutions like SSDs is almost always significantly faster than the total interaction time (prompt processing and generation time).
>
> While we agree that dense models excel in certain constrained settings, we mainly consider model inference in the scenarios listed above.

---

> ### Author Response · Authors · 2024-11-22
>
> **Comment**:  I generally prefer Gururangan et al., 2023's approach to evaluation, which pitches the MoE against a single larger but train-compute-matched dense model. Inference-time compute is matched by ensembling the predictions of the top k experts. I like it for the additional reason that it gestures toward the scenario where the dense model is so large that it does not fit on a single compute node and requires model-parallel training (whereas individual experts don't).
>
> **Responses**:
>
> Thank you for your valuable feedback and for highlighting the evaluation approach of Gururangan et al., 2023 that would be a great alternative to larger training compute matched dense baseline that requires model-parallel training. As you noted, we have performed a similar comparison by evaluating our 335M model with 32 experts against a 1.3B dense baseline (see Appendix A.3, Table 3). In this setup, the training compute is roughly matched, and our model achieves comparable perplexity with significantly less inference cost—three times less than the dense model.
>
> We acknowledge that conducting additional evaluations following a methodology similar to Gururangan et al., 2023 would further strengthen our findings. However, due to constraints in time and computational resources, we are currently unable to extend our work with additional results. We extremely appreciate your suggestion and plan to explore this evaluation strategy in future research. Thank you again for your constructive feedback regarding evaluation strategy.
>
> **Comment**: Throughout, the paper also needs to do a better job positioning itself relative to Gururangan et al., 2023. The abstract is pretty representative; with the exception of the phrase "according to a short prefix" and the name of the method, the abstract of this paper could have been that for the prior paper, which differs from this one only in the choice of routing function. It needs to be clearer that this isn't the first paper to attempt asynchronous training using MoEs and that all the novelty lies in the routing.
>
> **Response**:
>
> Thank you for pointing it out, we will rewrite the abstract for the revised version of the paper to make this clearer.
>
>
> **Question**: How were the number of tokens chosen for each configuration (see Weaknesses section above)?
>
> **Response**:
>
> Thank you for your question. We chose the number of training tokens based on recent studies ([1]–[3]) showing that training smaller models (e.g., 1.3B parameters) beyond the "optimal" number suggested by scaling laws ([4]) can improve test loss and downstream performance while maintaining low inference cost.
>
> As we mentioned in the answer above, our main reason for training the dense model on the same cumulative number of tokens as the experts was to demonstrate that, under the same inference budget, simply training a dense model for a longer time does not yield the same improvements in perplexity and downstream performance as scaling the number of experts does.
>
> [1] https://arxiv.org/pdf/2401.00448
>
> [2] https://arxiv.org/pdf/2302.13971
>
> [3] https://arxiv.org/pdf/2401.14196
>
> [4] https://arxiv.org/pdf/2203.15556

---

> > ### Comment · Reviewer_iShb · 2024-11-24
> >
> > Thanks for the detailed rebuttal! Excellent work---my concerns have been addressed.

---

> > > ### Author Response · Authors · 2024-11-25
> > >
> > > Dear Reviewer,
> > >
> > > Thank you for taking the time to review our rebuttal. We are glad to have addressed your concerns. Once again, we appreciate your detailed and helpful review, as well as your positive feedback.

---

### Meta-Review · Area_Chair_nea3 · 2024-12-10

**Metareview:**

This paper introduces a more parallelizable MoE system. The approach is very simple, introducing a single router to make asynchronous operation possible.

Pros: Appears to significant improve efficiency of an MoE. Very simple approach, can be further modified.

Cons: Longer prefix length makes it harder to reliably select the right expert. Baseline comparison is against a dense model with fewer total parameters than the entire mixture.

**Additional Comments On Reviewer Discussion:**

Reviewer iShb expressed skepticism about camping a single dense model to a mixture of the same size of models, since the MoE has extra weights inherently. The authors unfortunately do not have resources for the larger baseline, but answered some related questions based on results in the appendix.

---

### Decision · Program_Chairs · 2025-01-22

Accept (Spotlight)